# A Nationwide Seroprevalence Study for Measles in Individuals of Fertile Age in Romania

**DOI:** 10.3390/antib14020032

**Published:** 2025-04-02

**Authors:** Aurora Stanescu, Simona Maria Ruta, Mihaela Leustean, Ionel Iosif, Camelia Sultana, Anca Maria Panaitescu, Florentina Ligia Furtunescu, Costin Cernescu, Adriana Pistol

**Affiliations:** 1Faculty of Medicine, Carol Davila University of Medicine and Pharmacy, 020021 Bucharest, Romania; aurora.stanescu@insp.gov.ro (A.S.); madalina.sultana@umfcd.ro (C.S.); anca.panaitescu@umfcd.ro (A.M.P.); florentina.furtunescu@umfcd.ro (F.L.F.); adriana.pistol@umfcd.ro (A.P.); 2National Institute of Public Health, National Centre for Communicable Diseases Surveillance and Control, 050463 Bucharest, Romania; mihaela.leustean@insp.gov.ro (M.L.); ionel.iosif@insp.gov.ro (I.I.); 3Department of Emerging Viral Diseases, Stefan. S. Nicolau Institute of Virology, 030304 Bucharest, Romania; 4Filantropia Hospital, 011171 Bucharest, Romania; 5Medical Sciences Section, Romanian Academy, 010071 Bucharest, Romania; cernescucostineugen@gmail.com

**Keywords:** seroprevalence, measles, Romania, measles IgG antibodies, pregnant women

## Abstract

**Background/Objectives:** Romania remains endemic for measles due to suboptimal vaccine coverage rates. During the last three epidemics, the highest incidence of measles was recorded in children younger than 1 year, who should have been partially protected by maternal antibodies. A nationwide cross-sectional seroprevalence study was conducted on persons of fertile age, to evaluate potential immunity gaps in the population. **Methods:** Between June and October 2020, 959 serum samples were collected from individuals aged 25–44 years (46.5% females) from all the geographic regions in Romania. Measles IgG antibodies were assessed using an enzyme-linked immune assay (DIA.PRO-Diagnostic Bioprobes Srl, Italy). Statistical analysis was performed in IBM SPSS Statistics 27.0, using Fisher’s exact and chi-squared tests to test for associations between seropositivity and demographic factors, with *p* < 0.05 considered statistically significant. **Results:** The overall measles seroprevalence was 77%, without gender- or geographic region-related differences. Both the seropositivity rate and the measles antibodies titers increased with age, with the highest difference between the oldest and the youngest age group (*p* = 0.057), suggesting persistent immunity after natural infection in older individuals or anamnestic responses in vaccinated persons, caused by repeated exposures to the circulating virus. An additional confirmatory pilot study on 444 pregnant women confirmed the low level of measles seroprevalence (68.4%), with a significant upward trend in older ages (75% in those aged >40 years old vs. 65% in those aged 25–29 years, *p* = 0.018 and mean reactivity of measles antibodies 3.05 ± 1.75 in those aged >40 years vs. 2.28 ± 1.39 in those aged 25–29 years, *p* = 0.037). **Conclusions:** This study signals critical immunity gaps in the population that contribute to the accumulation of susceptible individuals and recurrent measles outbreaks. The absence of measles antibodies in women of childbearing age increases the newborn’s susceptibility to infection, with potentially severe complications.

## 1. Introduction

Measles is one of the most contagious human diseases, with an estimated reproduction rate in a susceptible population (R0) of 9–18, surpassing all other viral diseases (e.g., chickenpox R0 = 5–7, polio R0 = 4–13) [1]. As such, a high vaccination coverage (>97% for both doses of the measles vaccine and 95% measles immunity in children aged 1−9 years) is required to control virus transmission and to achieve the WHO elimination target [2,3,4]. Suboptimal vaccination coverage in specific regions and certain population groups, amplified by missed vaccine doses during the COVID-19 pandemic and by aggressive anti-vaccination campaigns, together with the increased population mobility, intensified by conflicts, war, and climatic and economic displacements, has led to measles re-emergence worldwide [5].

In Europe, over 95,000 measles cases were reported in 2024 by the WHO Regional Office. A significant increase in measles cases has already been reported in 2023 [6] as a consequence of decreasing vaccine coverage rates during the COVID-19 pandemic, with six Central Eastern European countries experiencing significant epidemics: Kazakhstan, Azerbaijan, Kyrgyzstan, Romania, the Russian Federation, and Türkiye. Kazakhstan and Kyrgyzstan had the highest incidence in the age group 1–4 years. In comparison, the Russian Federation and Türkiye reported the most significant proportion of measles cases in adults aged 20 years and older [7]. Apart from Türkiye, all the previously affected areas remain endemic for measles in 2024: Kazakhstan (27,760 cases), Romania (21,738 cases), Azerbaijan (16,675 cases), Russian Federation (14,751 cases), Kyrgyzstan (12,587 cases) [8]. In the EU/EEA, Romania has the highest notification rate and the highest mortality rate, mostly in children less than 5 years old and unvaccinated [8]. These epidemics arise because of persistent cumulative immunity gaps in the population caused by suboptimal vaccination or waning immunity in countries with good immunization programs [9]. To detect these immunity gaps and to effectively control the spread of measles, seroepidemiological prevalence studies must be conducted as part of the verification process for disease elimination [10,11,12,13].

In Europe, the last seroprevalence study for measles virus infection was conducted between 1996 and 2004 by the European Seroepidemiology Network (ESEN 2) as part of a more extensive study on age-specific seroprevalence for eight vaccine-preventable diseases (measles, mumps, rubella, diphtheria, pertussis, varicella-zoster, hepatitis A, hepatitis B) [14]. The results showed that Romania was one of the seven countries considered at risk of measles epidemics due to high susceptibility in children [15]. Indeed, during the last 15 years, Romania has experienced three large-scale measles outbreaks: between 2011 and 2012 (12,234 confirmed cases and three deaths), between 2016 and 2020 (over 20,000 confirmed cases and 64 deaths), and between 2023 and 2024 (over 25,000 cases and 22 deaths, still ongoing) [16,17,18]. The susceptible population has been continuously amplified due to the suboptimal coverage rates with both doses of the measles-mumps-rubella vaccine registered during the last 10–15 years [19]. The most significant number of cases and deaths in all three epidemics have been recorded in children up to one year of age, who should have been partially protected by maternal antibodies transmitted transplacentally and through breastfeeding. In this context, this study aimed to assess the seroprevalence of measles antibodies in Romania in persons of fertile age to evaluate potential immunity gaps that leave newborns unprotected against measles and to inform specific targeted vaccination campaigns.

## 2. Materials and Methods

**Subjects. (1) A nationwide cross-sectional study of individuals of fertile age was** conducted in persons 25–44 years old, the age groups in which the highest birth rate is recorded in Romania, according to the Eurostat data [20]. The sample size was calculated using a validated sampling strategy [21] to ensure national representativeness. A sample size of 959 subjects was established as statistically significant using the following formula in the EpiInfo 7 Program: n = [DEFF × Np(1 − p)]/[(d2/Z21 − α/2 × (N − 1) + p × (1 − p)]—where n represents the sample size, DEFF—the design effect for cluster surveys (1); N—the population size; p—the probability (50%); d—the error proportion (0.05), z—factor based on the confidence level (Z = 1.96 for a 95 percent confidence level) and 1 − α—the level of confidence (95%). The study population was divided by gender, four age groups: 25–29 years, 30–34 years, 35–39 years, and 40–44 years, and all 42 counties in Romania, considering the proportionality criterion, according to the demographic data published by the National Institute of Statistics [22]. We used for our selection a bank of serum samples collected between July and October 2020, within a seroprevalence study initially designed for COVID-19, and stored at −80 °C in the National Institute for Public Health (NIPH) laboratory, randomly selecting samples to ensure proportionality per gender, age-group, and county. The study was performed on completely anonymized, randomly selected remaining samples, for which informed consent for research use was previously obtained from individual patients. The study was approved by the Ethics Commission of the NIPH (Nr. 10056/22 May 2023).

**(2) Prospective pilot study on pregnant women.** To further confirm the seroprevalence results, a prospective pilot study was conducted in 444 pregnant women aged 25–45 years, randomly selected from those who required routine monitoring during pregnancy in a tertiary hospital in Bucharest between 2022 and 2023 and agreed to be enrolled in our study. The study was approved by the Ethics Committee of Clinical Hospital Filantropia, Bucharest (Nr. 75/03 August 2022).

**Anti-measles IgG antibody testing.** For both studies, measles IgG antibodies were tested using an enzyme-linked immune assay (Measles Virus IgG, DIA.PRO-Diagnostic Bioprobes Srl, Italy), with results expressed as reactivity (a ratio between the 450 nm optical density of the sample and the cut-off). According to the manufacturer protocol, the results were considered negative if <1 and positive if ≥1; the assay sensitivity is >98% and its specificity > 98%, calculated on a panel of positive samples derived from children who had clinical signs of measles infection and vaccinated children/adults and on a panel of samples derived from people with no clinical signs of measles and who were not vaccinated.

**Statistical analysis** was performed using IBM SPSS Statistics 27.0 for Windows, utilizing the Percentage Difference Calculation Test (Z) and corresponding confidence intervals of 95% (95% CI), with *p* < 0.05 considered statistically significant. Stratified analysis of the measles seroprevalence by age groups and by regions vs. national was performed in STATA/MP13.1 for Windows using the Percentage Difference Calculation Test (Z) and corresponding confidence intervals of 95% (95% CI), with *p* < 0.05 considered statistically significant.

## 3. Results

**Measles seroprevalence in individuals of fertile age.** The mean age of the 959 enrolled individuals was 35.2 ± 0.18 years, and 46.5% (n = 446) were females. There was a balanced distribution of age groups, genders, and geographic regions (Table 1). The overall seroprevalence of IgG anti-measles antibodies was 77%, and the measles seroprevalence rate increases with age, with the highest rates of seropositivity (83.3%) achieved in the age group 40–44 years. Further analysis (for corresponding confidence intervals of 95% estimated, with *p* < 0.05 considered statistically significant) did not reveal significant gender differences overall (*p* = 0.796) or within age group seroprevalence (Table 1).

We performed chi-square trends analysis to demonstrate that the frequency of seropositive samples increases with age and the results revealed that the measles seroprevalence rates differ statistically significantly between the age groups 25–29 years and 35–39 years (66.3% vs. 81.9% %, *p* = 0.0001) and also between the age groups 25–29 years and 40–44 years (66.3% vs. 83.3%, *p* = 0.0001) Table 2.

Distribution of seropositive and seronegative values of measles IgG antibodies by age group is suggestively represented in Figure 1. Considering that these values are of the order of units (between 0 and 10), we used the logarithmic scale for a good resolution and a readable graphical representation.

There are no statistically significant differences in the measles seroprevalence rates by geographic region, nor in the seroprevalence rates of different age groups from the different areas compared to the national seroprevalence.

The analysis of geographical distribution of mean reactivity rates of measles IgG antibodies showed that the northwest, northeast, southeast, and some of the central regions have the lowest values for the age groups selected in our study (Figure 2).

**Analysis of measles antibody reactivity.** Using reactivities as a surrogate for measles antibody titers, we analyzed the mean values between the age groups included in the study. Using the Dunnett t-test (treat one group as a control and compare all other groups against it) and corresponding confidence intervals of 95%, with *p* < 0.05 considered statistically significant, the highest difference is registered between the youngest age group studied (25–29 years) and the oldest age group (40–44 years), with marginal statistical significance. For the other age groups, there are no statistically significant differences in the reactivity values (Table 3).

For measles seropositive samples, the reactivities increase with age with a mean of 4.28 ± 0.32 in those aged 25–29, followed by the 30–34 y age group (4.93 ± 0.52), the 35–39 y age group (4.78 ± 0.24) and the 40–44 y age group (5.40 ± 0.28). A significant difference was noticed between the mean of 4.28 ± 0.32 in those aged 25–29 and the mean of 5.40 ± 0.28 in those aged 40–44 (*p* = 0.057).

**Measles seroprevalence in pregnant women.** The mean age of the enrolled pregnant women was 31.5 ± 5.7 years, and the mean gestational age was 21.8 ± 9.8 weeks. Only 68.4% of pregnant women had IgG measles antibodies, with a mean reactivity value of 2595 ± 1438. There is no significant difference between the seropositive and seronegative women according to the mean age (31.8 ± 0.3 years vs. 31.2 ± 0.3 years, *p* = 0.22) or urban/rural residence (70.5% vs. 67.7%, *p* = 0.3). The seropositivity rate is significantly higher for women aged > 40 years (75% vs. 65% in younger women, *p* = 0.018). The mean reactivity of measles antibodies in pregnant women has a significant upward trend by age groups (from 2.28 ± 1.39 [95% CI 1.98–2.56] in those aged 25–29 years to 3.05 ± 1.75 [95% CI 2.38–3.75] in those aged > 40 years, *p* = 0.037), with *p* < 0.05 considered statistically significant and the corresponding 95% confidence intervals (95% CI) estimated (Figure 3).

## 4. Discussion

According to the results of the ESEN 2 seroprevalence study [14] in Romania, in 2002, the rate of measles seronegative adults was 1.4% for the age group 20–39 years and 0.3% for those aged 40+ years. In the present study, we showed a marked increase, up to 23%, in the rate of measles seronegative individuals aged 25–45 years. All these individuals, except those aged 40–44 years, should have been vaccinated, as the measles vaccine was introduced in the national immunization program in 1979 in Romania. Although we did not have access to the vaccination records, the low measles seropositivity can be due to (a) lack of vaccination; (b) lack of seroconversion after vaccination due to primary vaccine failure [23], or breaches in the cold chain during vaccine transport or storage [24]; and (c) waning of measles antibodies [25]. A lower seroconversion rate after vaccination could be an explanation for individuals aged 30–39 years, representing the 1981–1990 birth cohorts, who should have received a single dose of measles vaccine administered at the age of 9–11 months, following the national vaccination schedule valid in Romania from 1979 to 1995. Vaccination at a younger age is less immunogenic, and measles vaccine efficacy in infants younger than 9 months was estimated to be 58% compared to 83% in those 9 months or older [26]. Several studies have also reported the waning of measles antibodies years after vaccination [27,28,29,30,31,32], and the antibody level after vaccination is often lower than that induced by natural measles virus infection [33,34,35]. Mathematical modeling has suggested that the waning of vaccine-induced immunity might be compatible with the recent trends in measles cases in England, with the rising number of breakthrough infections among individuals who have received two doses of the measles vaccine [36] and in Germany [37]. Nevertheless, in our cohort, the seropositivity rate and the measles antibody titers are lower in the youngest age group compared to the older age groups, making waning a less probable hypothesis [38].

Instead, in our measles endemic setting, the higher titers of measles antibodies in older individuals suggest life-long immunity following measles infection (most probable for the age group 40–44 years, born before the introduction of measles vaccination) or multiple anamnestic responses in vaccinated individuals, with vaccine-induced antibodies boosted by repeated exposures to the circulating virus. Additionally, we did not detect gender-related differences in the measles seropositivity rate or antibody titers, although vaccine-induced immune responses might differ between males and females [39], supporting hybrid immunity or gained after natural infection. The low seroprevalence seen in this study in young individuals (both in the nationwide study on those of fertile age and in the pilot study on pregnant women) probably reflects the continuously decreasing vaccine coverage rates reported in Romania [19]. These individuals, aged 25–29 years, should have received a complete measles vaccination regimen, as they were born after the introduction in 1995 of a second routine measles vaccine dose in the national immunization program.

The lack of humoral immunity seen in an essential part of this study’s subjects does not signify a lack of measles protection, as an anamnestic response might be developed after viral exposure, and cellular immunity might still be present and efficient, as reported for agammaglobulinemia patients who can be cured of measles [25]. Indeed, during the current measles epidemic in Romania, only 10.6% of all cases have been reported in adults, of which 33.9% had an unknown vaccination status, and only 2.9% and 2.8%, respectively, were vaccinated with one and two doses. Nevertheless, the lack of protective antibodies in young, active individuals of both genders increases the risk of viral spread in the community, workplaces, nurseries, kindergartens, and schools [40]. Frequent migration in and outside the country can amplify the transmission chains and explain the repeated measles outbreaks in Romania, fueled by the low vaccine coverage rates. Immunity gaps have been reported in other European countries [41,42,43,44], but at lower levels than the ones found in Romania, and measles outbreaks have been infrequent and with a limited number of cases. In addition, the absence of measles antibodies in women of fertile age means that there is no passive transplacental antibody transfer during future pregnancies, leaving the newborn completely unprotected against measles during the first year of life before reaching the recommended vaccination age [45,46].

The low rate of measles seropositivity found in the general population of fertile age is confirmed in the tested pregnant women and is further reflected by the high measles incidence in newborns aged 6 months or younger during the currently ongoing epidemic in Romania. As such, we suggest that the prenatal analysis package in Romania should also include testing for measles antibodies. This will allow vaccination of the seronegative women before conception or, in the case of already pregnant women, vaccination after delivery to ensure adequate protection for future pregnancies. Several studies have discussed the limits of EIA tests in seroprevalence analysis, highlighting a lower performance of EIA compared to plaque reduction neutralization tests (PRNT) [47], leading to potential misclassification of those with very low levels of measles antibodies as seronegatives. Even if this had been the case, a very low level of maternal antibodies would not be protective for the newborn. Vaccination of future mothers who are EIA seronegative will ensure a high level of antibodies during pregnancy and increase protection in newborns [48,49] who are at risk for severe measles complications [50]. As both vaccination-acquired maternal antibodies and passively transmitted antibody titers decline rapidly, an early vaccination of very young infants born to seronegative mothers might also be recommended during measles outbreaks [51,52]. A meta-analysis of the vaccine efficacy showed that vaccination before 9 months yields lower antibody titers compared to the regular administration at 9 or 12 months and that additional vaccine doses ensure a high seropositivity and cellular immunity, supporting early vaccination during outbreaks [26]. It is also tempting to speculate that once aware of the risk posed to the newborn by the absence of maternal antibodies, a mother will also acknowledge the need for all recommended childhood vaccinations, increasing vaccine acceptance and improving the national vaccine coverage rate. The maternal-fetal passive transferred humoral immunity can also influence the dynamics of pathogen transmission in the community and the epidemic patterns, as suggested in a mathematical model [53].

## 5. Conclusions

The results of this nationwide measles seroprevalence study reveal that an important proportion of individuals of fertile age in Romania are seronegative for measles antibodies, a fact further confirmed in a pilot study in pregnant women. Given the virus’s circulation in the community and the low vaccination coverage rate in the general population, this implies an increased risk of measles for newborns. To close these immunization gaps in the community, active immunization of seronegative women before or after pregnancy and early vaccination of susceptible infants must be deployed.

## Figures and Tables

**Figure 1 antibodies-14-00032-f001:**
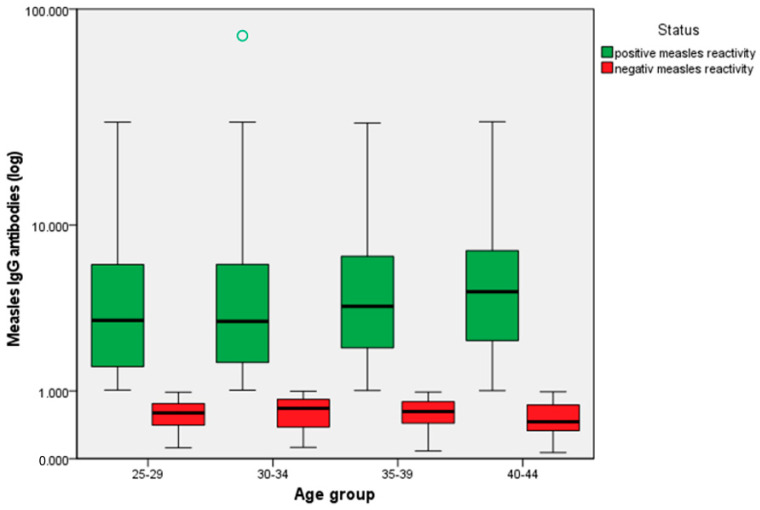
Distribution of seropositive and seronegative values of measles IgG antibodies by age groups. The black line represents the mean.

**Figure 2 antibodies-14-00032-f002:**
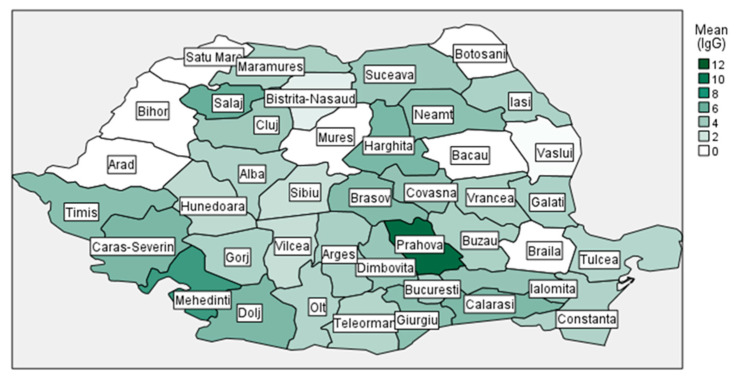
Geographical distribution of mean reactivity rates of measles IgG antibodies.

**Figure 3 antibodies-14-00032-f003:**
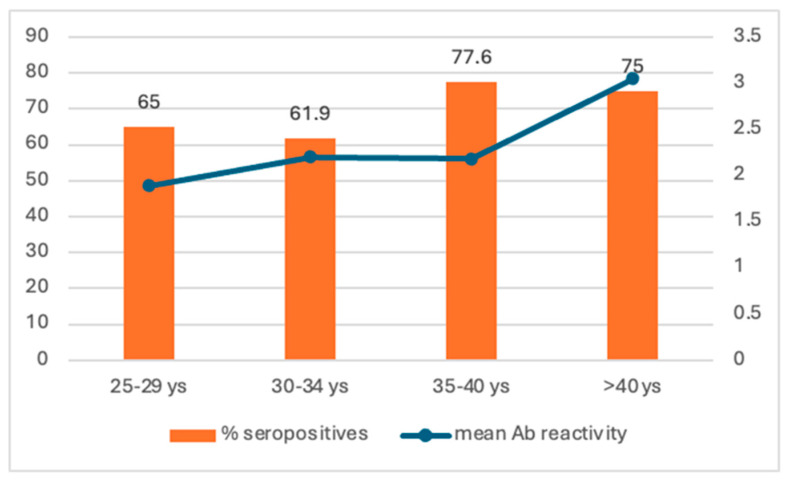
Seropositivity rate and mean reactivity of measles antibodies in pregnant women.

**Table 1 antibodies-14-00032-t001:** Seroprevalence of IgG anti-measles antibody values by age groups and gender.

Age Group	Females	Males	Seroprevalence of IgG Positives (%)	Females IgG Positives	Males IgG Positives	*p*
25–29 years (N = 193)	90	103	128 (66.3%)	55 (61.1%)	73 (70.9%)	0.153
30–34 years (N = 241)	112	129	176 (73%)	88 (78.6%)	88 (68.2%)	0.708
35–40 years (N = 243)	112	131	199 (81.9%)	90 (80.4%)	109 (83.2%)	0.565
40–44 years (N = 282)	132	150	235 (83.3%)	112 (84.8%)	123 (82%)	0.521
Total (N = 959)	446 (46.5%)	513 (54.5%)	738 (77%)	345 (77.4%)	393 (76.6%)	0.796

**Table 2 antibodies-14-00032-t002:** Chi-square test on seroprevalence of IgG anti-measles antibodies values by age groups.

	Age Group				Total
	25–29 y	30–34 y	35–39 y	40–44 y	
Nr of seropositives	128_a_	176_a,b_	199_b,c_	235_c_	738
Seroprevalence	66.30%	73.00%	81.90%	83.30%	77.00%

Each subscript letter denotes a subset of age group categories whose column proportions do not differ significantly from each other at the 0.05 level.

**Table 3 antibodies-14-00032-t003:** Comparative analysis of the measles reactivity rates between the age groups.

Age Group Compared	Age Group to Be Compared with	Mean Measles Reactivity Differences Between Age Groups	*p*
25–29 y	30–34 y	−0.656	0.870 (−2.293; 0.979)
	35–39 y	−0.505	0.772 (−1.591; 0.580)
	40–44 y	−1.127	0.057 (−2.275; 0.020)
30–34 y	35–39 y	0.151	1 (−1.386; 1.689)
	40–44 y	−0.470	0.966 (−2.052; 1.112)
35–39 y	40–44 y	−0.621	0.471 (−1.621; 0.377)

## Data Availability

All data presented are available upon request from first author (A.S.).

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
