# Peer review of "A Nationwide Seroprevalence Study for Measles in Individuals of Fertile Age in Romania"

_2073-4468, 2025, doi:10.3390/antib14020032_

Round 1
Reviewer 1 Report
Comments and Suggestions for Authors
Estimated Authors,
I've read with great interest the present, concise, but well written paper about the seroprevalence for measles in Romanian people.
The study has been performed through an appropriate methodology and, despite the limited number of sampled individuals, provides interesting information on this specific topic. In particular, Authors are able to suggest that seroprevalence rate for measles in fertile-age women is particularly low, providing a base of evidence for the high figures of measles outbreaks in Romania during the last decade.
From my point of view, this paper could be accepted nearly as it is, only minor adjustments are suggested and can be performed quite easily:
1) provide 95% confidence intervals for figure 1
2) the same for mean Ab reactivity for Figure 2
3) provide some further information about the sampling strategy: how where the individuals reached and actually sampled?
4) how representative may be considered the sample on pregnant women? please provide some comment in the discussion section
Author Response
Response for reviewer 1
We are grateful for the constructive review; we consider that the comments and suggestions had helped improved the quality of our article. We tried to clarify the following point:
Reviewer 1
Comments and Suggestions for Authors
Estimated Authors,
I've read with great interest the present, concise, but well written paper about the seroprevalence for measles in Romanian people.
The study has been performed through an appropriate methodology and, despite the limited number of sampled individuals, provides interesting information on this specific topic. In particular, Authors are able to suggest that seroprevalence rate for measles in fertile-age women is particularly low, providing a base of evidence for the high figures of measles outbreaks in Romania during the last decade.
From my point of view, this paper could be accepted nearly as it is, only minor adjustments are suggested and can be performed quite easily:
1) provide 95% confidence intervals for figure 1
We reconsidered Figure 1 and we explained the data in text at lines 169-171
2) the same for mean Ab reactivity for Figure 2
We added 95% confidence intervals at lines 182-184
3) provide some further information about the sampling strategy: how where the individuals reached and actually sampled?
The samples consisted of residual sera collected between July and October 2020, within a seroprevalence study initially designed for COVID-19, using a convenience sampling method from people of all age groups who presented themselves conjecturally for usual blood analysis collections at laboratories with high addressability, serving ambulatory patients (non-hospitalized). Each of the 42 County Public Health Directorates in Romania selected between 3 and 5 laboratories to participate in the study, only individuals who expressed their informed consent were enrolled. The sample size was calculated using the EpiInfo 7 program, to be representative regionally and on decadal age-group. Residual samples were stored at -80C in the laboratory of the National Institute for Public Health (NIPH). For this study, the required number of samples per gender, age-group and county were randomly selected, from the stored sera, with sample size determined according to the methodology described in the methods section.
A sentence was added in the manuscript at lines 87, 93 – 99.
4) how representative may be considered the sample on pregnant women? please provide some comment in the discussion section
The sample of pregnant women was constructed upon the feasibility criterion, by enrolling pregnant women who were routinely monitored during pregnancy in a tertiary center, and agreed to take part to the study. This limits the national representativity of our results from the statistical perspective. Although this is only a pilot study, the results confirm the national seroprevalence in fertile age women, and there are no premises to believe that other pregnant women have a better seroprevalence status. We are planning to carry out a study with national representativeness in the near future.
A sentence was added in the manuscript at lines 106 -107.

Reviewer 2 Report
Comments and Suggestions for Authors
This is a good manuscript dealing with the population immunity against measles in an affected country. The goal was to understand how young babies are not protected by maternal IgG . Nice idea.
The problems are in the text.
The authors must present the validation data of their serology assay. The cutoff are a commercial limit proposed by the producers and could be biased. For example, if they are interested in attribute to the mother antibodies the absence of the protection of the babe, they suggest a higher cut off values for the assay for more negative mothers. You must look the the specificity and sensitivity of the assay. This difference in limits could explain easily the difference of prevalence of your data and the previous published data. 20% to 2% is a large difference but easily explained by biased limits. If You have data, present the dot plot of each age group for seeing the difference between negative and positives.
Table 1 and 2 use the same distribution and data and table 1 had not significant results. The tables could be easily mixed without problems.
Figure 1 had several problems. The caption is so small that do not allow You to understand which data is presented. It is necessary?
Table 3 is incomprehensible, with incomplete analysis . If You want to compare prevalence os measles antibodies in different romanian regions is better to use the tables complete with crossed comparisions. All vs all. Use only one age group, probably the younger is the most interesting. Older groups are more similar.
Your idea is excellent, and probably could be easily better explained by using paired umbilical cord blood from normal babies and mother serum.
Comments on the Quality of English Language
The data is good, the text is intelligent but need revision
Author Response
Response for reviewer 2
We are grateful for the constructive review; we consider that the comments and suggestions had helped improved the quality of our article. We tried to clarify the following point:
Reviewer 2
Comments and Suggestions for Authors
This is a good manuscript dealing with the population immunity against measles in an affected country. The goal was to understand how young babies are not protected by maternal IgG . Nice idea. The problems are in the text. The authors must present the validation data of their serology assay. The cutoff are a commercial limit proposed by the producers and could be biased. For example, if they are interested in attribute to the mother antibodies the absence of the protection of the babe, they suggest a higher cut off values for the assay for more negative mothers. You must look the specificity and sensitivity of the assay. This difference in limits could explain easily the difference of prevalence of your data and the previous published data. 20% to 2% is a large difference but easily explained by biased limits. If You have data, present the dot plot of each age group for seeing the difference between negative and positives.
We have used Measles Virus IgG, DIA. PRO-Diagnostic Bioprobes Srl, Italy, a commercial kit, that allows for a semi-quantitative determination of IgG antibodies to Measles Virus in human plasma and sera. The assay results are expressed as reactivity (a ratio between the 450 nm optical density of the sample and the cut-off, where the cut-off represents the mean OD450nm value of the Negative Control (NC) + 0.250. The results are reported as negative if < 1 and positive if ≥ 1, a standard way of expressing results for a qualitative immunenzymatic assays (EIA). The kit is intended mostly for the follow-up of measles virus vaccination and the follow-up of infected individuals and is used internationally for both diagnostic and immunity studies. The assay is not quantitative and does not propose a threshold of maternal antibodies to define the presence/absence of passive protection for the newborns. According to the manufacturer, the assay sensitivity is >98% and its specificity > 98%, calculated on a panel of positive samples derived from children who had clinical signs of measles infection and vaccinated children/adults and on a panel of samples derived from people with no clinical signs of measles and who were not vaccinated.
Several studies have discussed the limits of EIA tests in seroprevalence analysis, highlighting a lower performance of EIA compared to PRNT (Latner DR, et al. J Clin Microbiol. 2020;58(6):e00265–e320. doi: 10.1128/JCM.00265-20;Lutz CS, et al BMC Infect Dis. 2023 May 31;23(1):367. doi: 10.1186/s12879-023-08199-8). We acknowledge that the results would have benefited from a confirmation using the more specific plaque reduction neutralization tests (PRNT). Nevertheless, this assay is technically difficult and resource-consuming for seroprevalence studies. Although EIA results are less sensitive than PRNT and may underestimate the level of immunity, we consider that low levels of maternal antibodies will not adequately protect the newborns in an endemic setting and that immunization of fertile age women may represent an efficient way of improving vaccine coverage rates in a country with periodic measles outbreaks.
A sentence was added in the manuscript at lines 114-118; 249-252.
Following reviewer 2’s recommendations we performed a graphical representation and the result is Figure 1. Distribution of seropositive and seronegative values of measles IgG antibodies by age group available in manuscript at lines 141 – 144. A sentence was added in the manuscript at lines 137-140.
Table 1 and 2 use the same distribution and data and table 1 had not significant results. The tables could be easily mixed without problems.
Following reviewer 2’s recommendations we merged the two tables and the result is Table 1. Seroprevalence of IgG anti-measles antibodies values by age groups and gender available in manuscript at lines 135 – 136.
Figure 1 had several problems. The caption is so small that do not allow You to understand which data is presented. It is necessary?
We reconsidered Figure 1 and we explained the data by adding a sentence in manuscript at lines 169-171.
Table 3 is incomprehensible, with incomplete analysis . If You want to compare prevalence os measles antibodies in different romanian regions is better to use the tables complete with crossed comparisions. All vs all. Use only one age group, probably the younger is the most interesting. Older groups are more similar.
Following reviewer 2’s recommendations we reconsidered Table 3 which is now Table 2 Comparative analysis of measles seroprevalence values - regions versus national for age group 25-29 y available in manuscript at lines 152-154. A sentences were added in manuscript at lines 146 -151; 155-156.
Your idea is excellent, and probably could be easily better explained by using paired umbilical cord blood from normal babies and mother serum.
We agree this will be an ideal way to prove the absence of transplacental transferred antibodies. Unfortunately, there were no adequate paired samples from mothers and babies. We are planning to collect such samples in future prospective studies. However, for the present study, we consider that the high number of newborn measles cases during all recent outbreaks is a reliable, though indirect proof for the absence of maternal transferred protection.
Comments on the Quality of English Language
The data is good, the text is intelligent but need revision
We have revised the entire text and made the required changes.

Round 2
Reviewer 2 Report
Comments and Suggestions for Authors
The authors performed several improvements in the text and methods presentation, but they had some difficulties presenting their data.
The statistical analysis was not as deep as necessary. Age group effect is easily desmonstrable, but their option for comparing each value with each other is not a good idea, due to freedom degree losses. The frequency of seropositive samples increases with age and is more easily demonstrable by Chi-Square trends analysis, which could be performed in both table 1 and Figure 2.
(I checked their data from Table 1 in my system and the significance was high, with p<0.001).
As a suggestion for table 2, the geographic distribution of seropositivity must be included in a map, with each population and vaccine coverage if possible, demonstrating that frontier areas could have the worst vaccine coverage, and justifying the inclusion in the manuscript.
The importance of antenatal vaccination is clear and those modifications are minor changes in the manuscript.
Author Response
Response for reviewer 2
We are grateful for the constructive review; we consider that the comments and suggestions had helped improved the quality of our article. We tried to clarify the following point:
Reviewer 2
Comments and Suggestions for Authors
The authors performed several improvements in the text and methods presentation, but they had some difficulties presenting their data.
(I checked their data from Table 1 in my system and the significance was high, with p<0.001).
We rephrased the explanation of data presented in the Table 1.
A sentence was added in the manuscript at lines 129 - 134.
The statistical analysis was not as deep as necessary. Age group effect is easily desmonstrable, but their option for comparing each value with each other is not a good idea, due to freedom degree losses. The frequency of seropositive samples increases with age and is more easily demonstrable by Chi-Square trends analysis, which could be performed in both table 1 and Figure 2.
Following reviewer 2’s recommendations we performed Chi-Square trends analysis to demonstrate that the frequency of seropositive samples increases with age and the result is Table 2. Chi-Square test on seroprevalence of IgG anti-measles antibodies values by age groups available in manuscript at lines 146-149. A sentence was added in the manuscript at lines 138-142.
As a suggestion for table 2, the geographic distribution of seropositivity must be included in a map, with each population and vaccine coverage if possible, demonstrating that frontier areas could have the worst vaccine coverage, and justifying the inclusion in the manuscript.
Following reviewer 2’s recommendations we performed a geographical distribution of mean reactivity rates of measles IgG antibodies and the result is Fig 2. Geographical distribution of mean reactivity rates of measles IgG antibodies available in manuscript at lines 166-167.
A sentence was added in the manuscript at lines 163-165.
The importance of antenatal vaccination is clear and those modifications are minor changes in the manuscript.
We performed the changes in the manuscript following reviewer 2’s recommendations.
